# Learning Temporal Higher-order Patterns to Detect Anomalous Brain Activity

**Ali Behrouz**
Cornell University
Ithaca, NY
ab2947@cornell.edu

**Farnoosh Hashemi**
Cornell University
Ithaca, NY
sh2574@cornell.edu

## Abstract

Due to recent advances in machine learning on graphs, representing the connections of the human brain as a network has become one of the most pervasive analytical paradigms. However, most existing graph machine learning-based methods suffer from a subset of five critical limitations: They are (1) designed for simple pairwise interactions while recent studies on the human brain show the existence of higher-order dependencies of brain regions, (2) designed to perform on pre-constructed networks from time-series data, which limits their generalizability, (3) designed for classifying brain networks, limiting their ability to reveal underlying patterns that might cause the symptoms of a disease or disorder, (4) designed for learning of static patterns, missing the dynamics of human brain activity, and (5) designed in supervised setting, relying their performance on the existence of labeled data. To address these limitations, we present HADIB, an end-to-end anomaly detection model that automatically learns the structure of the hypergraph representation of the brain from neuroimage data. HADIB uses a tetra-stage message-passing mechanism along with an attention mechanism that learns the importance of higher-order dependencies of brain regions. We further present a new adaptive hypergraph pooling to obtain brain-level representation, enabling HADIB to detect the neuroimage of people living with a specific disease or disorder. Our experiments on Parkinson's Disease, Attention Deficit Hyperactivity Disorder, and Autism Spectrum Disorder show the efficiency and effectiveness of our approaches in detecting anomalous brain activity.

## 1 Introduction

Recent advancements in neuroscience and neuroimaging have led researchers to shift from examining isolated brain regions to exploring network models [1]. This shift is largely attributed to the rapid progress in technologies like functional Magnetic Resonance Imaging (fMRI) and structural Magnetic Resonance Imaging (sMRI) [2], which can provide large-scale neuroimaging data with better quality. In network models of the brain, brain regions of interest (ROIs) are represented as nodes, and the similarities between these regions form edges [3]. Brain network models have demonstrated their effectiveness in enhancing our understanding of brain diseases and disorders [4, 5]. As a result, empirical data on brain networks has substantially increased in size and complexity, leading to a strong demand for appropriate tools and methods to model and analyze this data [5]

Simultaneously, machine learning techniques for analyzing graph-structured data have gained attention across various fields, including drug discovery, neuroscience, and biology [6–8]. Although numerous studies have confirmed the effectiveness of machine learning for human brain network analysis, most have concentrated on graph or node classification tasks [9, 10]. These tasks usually aim at disease detection [11], biological feature prediction [9], or functional system identification [12].

NeurIPS 2023 AI for Science Workshop.

However, the identification of abnormal brain activity, especially in those with neurological disorders, remains a critical focus for researchers. This is essential for understanding the mechanisms behind symptoms, enabling early detection, and facilitating the development of medical treatments. Most existing studies consider pairwise interaction among brain regions, neglecting the effect of non-pairwise interactions on the emerging dynamics. Recently, several studies have discussed the importance of higher-order correlation of brain regions. Rosenthal et al. [13] show that the network context might not be directly accessible at the level of individual regions, and Santoro et al. [14] show the signatures of higher-order patterns in brain functional activity. Temporal hypergraph representation of the brain from neuroimaging data can address this limitation by capturing both higher-order interactions of brain regions as well as higher-order patterns that are correlated to emerging dynamics of brain activity. Hypergraphs are powerful paradigms to model higher-order interactions, where each connection, also known as hyperedge, can connect a group of nodes at once.

**Limitation of Previous Methods.** The process of brain network analysis generally unfolds in two sequential steps. Initially, brain networks are derived from individual neuroimage data (e.g., fMRI). This typically involves choosing a brain atlas, which identifies specific ROIs to serve as nodes, and edges that show association between ROIs. For example in fMRI data, from each designated brain region, fMRI blood-oxygen-level-dependent (BOLD) signal sequences are then extracted, and the subsequent phase of edge generation involves calculating pairwise connectivity between these nodes, often based on Pearson correlation and/or mutual information. Finally, the connectivity measures established between node pairs are utilized for the downstream tasks. Although this process along with diverse machine learning methods is widely used in the existing literature [9, 15, 16], both the pipeline and existing methods suffer from a subset of the following limitations, which makes directly applying them in real-world scenarios challenging and/or impractical:

① Capturing linear correlation and ignoring temporal order: Most existing studies assume that the true dependency structure between brain regions is known prior to model training. That is, existing methods often use Pearson correlation and/or mutual information between the signals of brain regions (e.g, BOLD signals), while there is no statistical measure of dependency for truly capturing functional connectivity [17]. Moreover, these statistical correlations focus on capturing linear correlation and ignore temporal order, which means shuffling the time in each time window does not change the results. In order to mitigate the above limitation and to ensure that the model can effectively learn meaningful network representations for use in downstream tasks it is crucial to establish an approach for constructing the dependency structure of the network that accurately reflects the underlying neuroimage data.

② Missing higher-order dependencies: Existing studies assume that the dependencies of brain regions can be captured by pair-wise interactions, while the signatures of higher-order patterns in brain functional activity have been seen in recent studies [14]. Moreover, the network context might not be directly accessible at the level of individual regions and requires considering higher-order interactions between a group of regions [13, 18]. Accordingly, ignoring the higher-order dependencies of ROIs might result in suboptimal performance.

③ Missing hierarchical structure of the brain: The human brain is comprised of functional systems (FS) [19], which are groups of ROIs that perform similar functions as an integrated network [20]. These communities are essential in understanding the functional organization of the brain [21] as ROIs in the same functional modules tend to have similar behaviors and clustered representations [20]. Most existing studies neglect the hierarchical structure of the brain, leading to missing the functional dependencies of ROIs in the same FS.

④ Missing the dynamics of the network: Some existing studies neglect the fact that the functional connectivity of the human brain dynamically changes over time, even in resting-state neuroimaging data [22]. In task-dependent neuroimage data, subjects are asked to perform different tasks in different time windows, and the dynamics of the brain activity during these tasks play an important role in understanding neurological disease/disorder [23]. Moreover, Liégeois et al. [24] shows that brain dynamics at different timescales capture distinct aspects. Accordingly, mitigating the above limitation requires a model that not only captures the dynamics of the brain activity but it also can capture its dynamics at different timescales.

⑤ Designed for classification tasks: Most existing studies have focused on semi-supervised/supervised classification tasks, e.g., detecting diseases [11], predicting biological features [9], or identifying functional systems [12]. This setting not only relies their performance on

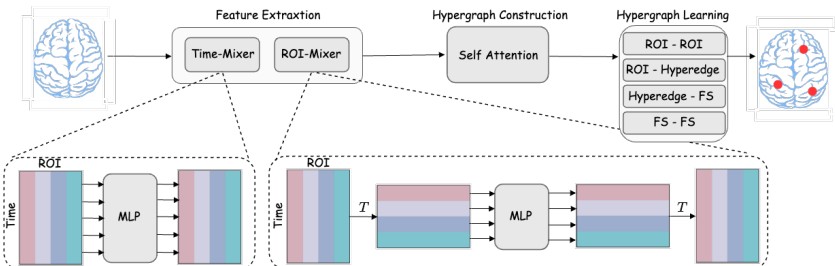

Figure 1: **Schematic of the HADɪB**. HADɪB consists of three stages: (1) Feature Extraction, (2) Hypergraph Construction, and (3) Hypergraph Learning.

the existence of labeled data, but it also limits their ability to reveal underlying patterns that might cause the symptoms of a disease or disorder. That is, a black-box classification method only predicts the label for a given brain network while understanding the cause of the brain disease/disorder (i.e., detecting abnormal brain activity) is a crucial step in facilitating early detection, and developing medical treatments.

**Contributions.** To address all the above limitations, we design HADɪB (**H**ypergraph **A**nomaly **D**etection **i**n **B**rain), an end-to-end unsupervised anomaly detection method that can detect anomalous patterns at different level of granularity. HADɪB first uses a novel hierarchical multivariate time-series encoder that can capture both cross-time and cross-ROI dependencies of ROIs' signals at different time scales. Next, to mitigate ① and ②, HADɪB employs a temporal hypergraph constructor that learns the temporal higher-order dependencies between ROIs' signals. To address the ③ and ④, HADɪB uses a tetra-stage message passing at different levels of granularity to learn the ROI, functional system, and brain encodings, taking advantage of the hierarchical structure of the brain. Experimental evaluation supports the need for considering higher-order patterns and shows the superior performance of HADɪB over baselines, as well as the importance of HADɪB's critical components. Finally, using real-world case studies we show how HADɪB can be used to detect abnormal brain regions or functional systems in a control group with brain disease or disorder.

## 2 Related Work

**Anomaly Detection in Hypergraphs and Time Series.** The problem of anomaly detection, which aims to detect abnormal nodes, edges, or subgraphs within a graph, has been extensively studied for both static and temporal graphs [25, 26]. Surprisingly, hypergraph anomaly detection is relatively unexplored. Park et al. [27] uses scan statistics to detect anomalous nodes. Leontjeva et al. [28] suggest taking advantage of the structural features of the nodes to detect anomalies. Recently, Lee et al. [29] designed a fast and effective algorithm based on a proximity matrix to detect abnormal nodes. Moreover, several studies use hypergraphs to learn higher-order patterns in time series forecasting [27, 30]. However, all these methods ① are designed for general applications and cannot take advantage of special properties of the brain, ② cannot detect anomalies at different levels of granularity (e.g., hyperedge, node, etc.).

**Machine Learning and Anomaly Detection in Brain Networks.** Several studies have recently analyzed brain networks to differentiate between healthy and diseased human brains [31–33]. With the success of graph neural networks in graph data analysis, deep learning models have been developed to predict brain diseases by studying brain network structures [34, 15, 35, 36, 11, 10]. However, these models focus on graph or node classification and aren't directly suited for anomaly detection. There are also methods aimed at detecting anomalies in brain regions or subgraphs, which could signal disease presence [4, 37, 38]. These studies adopt non-learning approaches and rely on pre-defined patterns or rules for detecting anomalies. Due to the complex nature of brain activity, it is unrealistic to assume that all abnormal brain activities follow the same pre-define pattern or rule. To address this limitation and to learn the abnormal patterns in the human brain from data, recently, Behrouz and Seltzer [39] proposed ADMɪRE, an unsupervised anomaly detection method that uses multiplex random walks to extract networks motif and to detect anomalous patterns in the brains of people living with a disease or disorder, accordingly.

All these methods target anomaly detection in brain networks with pairwise interactions, missing temporal higher-order patterns that play important roles in understanding the network context and its dynamics over time [13, 14]. Moreover, they do not take advantage of the hierarchical structure of the brain, missing the dependencies of ROIs' activity in the same functional systems [20, 21].

## 3 Methods

Each neuroimage data (e.g., fMRI or EEG) can be represented as a multivariate timeseries $\mathbf{X}_{1:T} = (\mathbf{x}_1, \dots, \mathbf{x}_T) \in \mathbb{R}^{n \times T}$, where $n$ is the number of ROIs. Next, we formally define temporal hypergraphs:

**Definition 1** (Temporal Hypergraphs). *A temporal hypergraph $\mathcal{G} = (\mathcal{V}, \mathcal{E}, \mathcal{X})$, can be represented as a sequence of snapshots $\mathcal{G} = \{\mathcal{G}^{(t)}\}_{t=1}^T$ that arrive over time, where $\mathcal{G}^{(t)} = (\mathcal{V}, \mathcal{E}^{(t)}, \mathcal{X}^{(t)})$ is the t-th snapshot, $\mathcal{V}$ is the set of nodes, $e_i \in \mathcal{E}^{(t)} \subseteq 2^{\mathcal{V}}$ are hyperedges, and $\mathcal{X}^{(t)} \in \mathbb{R}^{|\mathcal{V}| \times f}$ is a matrix that encodes node attribute information for nodes in $\mathcal{V}$. Note that we treat each hyperedge $e_i$ as the set of all vertices connected by $e_i$.*

Given a hypergraph $\mathcal{G} = (\mathcal{V}, \mathcal{E}, \mathcal{X})$, we represent the set of hyperedges attached to a node $u$ before time $t$ as $\mathcal{E}^{(t)}(u) = \{(e, t') | t' < t, u \in e\}$. We say two hyperedges $e$ and $e'$ are adjacent if $e \cap e' \neq \emptyset$ and use $\mathcal{E}^t(e) = \{(e', t') | t' < t, e' \cap e \neq \emptyset\}$ to represent the set of hyperedges adjacent to $e$ before time $t$. Each hypergraph snapshot $\mathcal{G}^{(t)}$ can be represented by an adjacency matrix $\mathcal{A}^{(t)} \in \{0, 1\}^{|\mathcal{V}| \times |\mathcal{E}^{(t)}|}$ such that $\mathcal{A}_{u,e}^{(t)} = 1$ if $u \in e$ and $\mathcal{A}_{u,e}^{(t)} = 0$ otherwise.

We focus on the problem of anomaly detection in the human brain at three levels of granularity: ① Hyperedge anomaly detection, ② ROI anomaly detection, and ③ FS/Brain anomaly detection.

### 3.1 HADIB: An Unsupervised Anomaly Detection Method in Human Brain

In this section, we design HADIB, an unsupervised anomaly detection method for the human brain, and discuss each of its components in detail. Figure 1 illustrates the overview of HADIB architecture.

**Hierarchical Feature Extraction from Time Series.** The Time Mixer module aims to encode the multivariate time series data at different time scales. To this end, we first map the input sequence $\mathbf{X}_{1:T}$ into a hierarchical series $\mathbb{X} = \{\mathbf{X}^{(s)}\}_{s \in \mathcal{S}}$ such that $\mathbf{X}_{1:T_s}^{(s)} = (\mathbf{x}_1^{(s)}, \dots, \mathbf{x}_{T_s}^{(s)}) \in \mathbb{R}^{n \times T_s}$, $\mathcal{S}$ is the set of all scales, and each $s \in \mathcal{S}$ is a time window of $\mathbf{X}^{(s)}$ (i.e., $T_s = \lceil \frac{T}{s} \rceil$). At each time scale $s \in \mathcal{S}$, we need to capture both cross-ROI and cross-time dependencies. Accordingly, inspired by MLP-MIXER [40], we use a Time-Mixer and ROI-Mixer modules to capture cross-time and cross-ROI dependencies, respectively. In the Time-Mixer module, we have:

$$\mathbf{H}_{\text{TIME}}^{(s)} = \mathbf{X}^{(s)} + \mathbf{W}_{\text{TIME}}^{(1)^{(s)}} \sigma \left( \mathbf{W}_{\text{TIME}}^{2^{(s)}} \texttt{LayerNorm} \left( \mathbf{X}^{(s)} \right) \right),$$

where $\mathbf{W}_{\text{TIME}}^{(1)^{(s)}}$ and $\mathbf{W}_{\text{TIME}}^{(2)^{(s)}}$ are learnable metrices, $\texttt{LayerNorm}(.)$ is layer-normalization [41], and $\sigma(.)$ is a non-linearity (e.g., GELU [42]). To capture cross-ROI dependencies, we use ROI-Mixer as follows:

$$\begin{aligned} \mathbf{H}_{\text{ROI}}^{(s)} = \mathbf{H}_{\text{TIME}}^{(s)} \\ + \sigma \left( \texttt{LayerNorm} \left( \mathbf{H}_{\text{TIME}}^{(s)} \right) \mathbf{W}_{\text{ROI}}^{(1)^{(s)}} \right) \mathbf{W}_{\text{ROI}}^{(2)^{(s)}}, \end{aligned} \tag{1}$$

where $\mathbf{W}_{\text{ROI}}^{(1)^{(s)}}$ and $\mathbf{W}_{\text{ROI}}^{(2)^{(s)}}$ are learnable metrics. Note that, while the feature extractor consists of simple all-MLP Time- and ROI-Mixer modules, its architecture (i.e., capturing cross-time and cross-ROI depenndencies) make it a powerful model (see § 4).

Finally, we project $\mathbf{H}_{\text{ROI}}^{(s)}$ to have the same size encoding for different time scales:

$$\mathbf{H}_{\text{PROJ}}^{(s)} = \text{MLP} \left( \mathbf{H}_{\text{ROI}}^{(s)} \right). \tag{2}$$

The above procedure encodes the signals at different time scales separately. Accordingly, it does not take advantage the complementary information provided by different time scales [24]. To address

this challenge, we concatinate the $\mathbf{H}_{\text{PROJ}}^{(s)}$ for different $s \in \mathcal{S}$ in the order of the time scale, and use an MLP-MIXER module [40] to combine the encodings:

$$\mathbf{H}_{\text{F}} = \text{MLP-MIXER}\left(\bigoplus_{s \in \mathcal{S}} \mathbf{H}_{\text{PROJ}}^{(s)}\right), \tag{3}$$

where $\bigoplus$ denotes concatination along the ROI dimension (i.e., $\mathbf{H}_{\text{F}} \in \mathbb{R}^{n \times d}$).

**Temporal Hypergraph Construction.** To capture dependencies between brain regions, while considering the extracted temporal features, we use a self-attention mechanism to learn temporal hypergraph representation of the brain, where each node is a ROI and each hyperedge represents the association of a group of nodes. As discussed in §1, the connectivity of the human brain dynamically changes over time [22]. For example in task-dependent fMRI, subjects are asked to perform a task in each time window. To this end, we use $s_{\max} = \max\{s \in \mathcal{S}\}$, as the time window to construct different snapshots of the network. Accordingly, using zero-padding, we use the above feature extraction procedure for $\mathbf{X}_{1:(t+1)s_{\max}}$ to obtain $\mathbf{H}_{\text{F}}^{(t)}$ for all $0 \leq t \leq \lfloor \frac{T}{s_{\max}} \rfloor$. Next, to construct the $t$-th snapshot of the hypergraph, we use modified self-attention mechanism [43] as follows:

$$\mathcal{A}^{(t)} = \text{SIGMOID}\left(\frac{Q^{(t)} K^{(t)}}{\sqrt{K^{(T)^\top}}}\right), \tag{4}$$

where $Q^{(t)} = \mathbf{H}_{\text{F}}^{(t)} \mathbf{W}_{\text{ATTN}}^{(1)}$, $K^{(t)} = \mathbf{H}^{(t)^\top}_{\text{F}} \mathbf{W}_{\text{ATTN}}^{(2)}$, and $\mathbf{W}_{\text{ATTN}}^{(1)}$ and $\mathbf{W}_{\text{ATTN}}^{(2)}$ are learnable matrices. We interpret $\mathcal{A}^{(t)}$ as the adjacency matrix of the hypergraph representation of the brain in the $t$-th snapshot. Inspired by results reported by Said et al. [44], we use $\mathcal{X}^{(t)} = \mathbf{H}_{\text{F}}^{(t)}$ as the node features.

**Hypergraph Message-Passing.** After constructing the temporal hypergraph, to learn the dependencies between temporal patterns of different scales, we propose a tetra-stage message passing mechanism, which contains ① ROI-ROI, ② ROI-Hyperedge, ③ Hyperedge-FS, and ④ FS-FS.

① ROI-ROI phase: To learn the local dependencies of ROIs, we iteratively aggregate messages from the local neighborhood of ROIs. Given $t$-th snapshot, if $u$ and $v$ are in a hyperedge $e \in \mathcal{E}^{(t)}$, then we update ROI encodings in $\ell$-th layer of neural network as follows:

$$m_{u \to v}^{(\ell)} = \mathbf{W}_{\text{ROI}}^{(\ell)} \text{CONCAT}\left(\hat{\mathbf{h}}_u^{(t)^{(\ell-1)}}, \hat{\mathbf{h}}_v^{(t)^{(\ell-1)}}, \zeta_e^{(t)^{(\ell-1)}}\right),$$
$$\hat{\mathbf{h}}_u^{(t)^{(\ell)}} = \text{SUM}\left(\left\{m_{v \to u}^{(\ell)} | v \in \mathcal{N}^{(t)}(u)\right\}\right) + \hat{\mathbf{h}}_u^{(t)^{(\ell-1)}}, \tag{5}$$

where $\mathbf{W}_{\text{ROI}}^{(\ell)}$ is a learnable matrix, $\zeta_e^{(t)^{(\ell-1)}}$ is the encoding (strength) of hyperedge $e$ (we discuss and compute it in ②), and $\mathcal{N}^{(t)}(u)$ is the set of nodes that are connected to $u$ by at least a hyperedge in $t$-th snapshot. We initialize the $\hat{\mathbf{h}}_u^{(t)^{(0)}} = \mathbf{h}_F^{(t)}(u)$, where $\mathbf{h}_F^{(t)}(u)$ is the corresponding row of matrix $\mathbf{H}_F^{(t)}$ for $u$.

② ROI-Hyperedge phase: To obtain the encoding of a hyperedge, one can simply consider the summation of all ROI's encoding connected by the hyperedge. However, the strength of each dependencies for each ROI is different. Accordingly, we use the attention mechanism in Equation 4 to learn the importance of each hyperedge for each node (ROI) in the hypergraph. Let $\zeta_e^{(t)}$ be the encoding of hyperedge $e \in \mathcal{E}^{(t)}$:

$$\zeta_e^{(t)^{(\ell)}} = \sum_{u \in e} \mathcal{A}_{u,e}^{(t)} \hat{\mathbf{h}}_u^{(t)^{(\ell)}}, \tag{6}$$

where $\mathcal{A}_{u,e}^{(t)}$ is computed by Equation 4.

③ Hyperedge-FS phase: We consider each FS as a set of ROIs. Accordingly, to encode an FS from its ROIs' encoding one might suggest aggregating the encoding of its ROIs. However, this aggregation misses the dependencies of ROIs in each FS. To this end, we propose a pooling function $\text{POOL}(.)$ that aggregates the encodings of hyperedge within each FS. This approach not only incorporates the information from different ROIs, but also considers their dependencies and their roles in the dynamics

of the system. Let $\mathcal{F}$ be the set of all functional systems, for $f \in \mathcal{F}$ we use $\mathcal{E}_f$ to denote the set of hyperedges within the $f$ and matrix $\mathbf{Z}_f^{(t)(\ell)} = \bigoplus_{e \in \mathcal{E}_f} \zeta_e^{(t)(\ell)}$. We compute the FS encoding as:

$$\hat{\mathbf{Z}}_f^{(t)(\ell)} = \text{GRU}\left(\mathbf{Z}_f^{(t)(\ell)}, \mathbf{Z}_f^{(t-1)(\ell)}\right)$$

$$\tilde{\mathbf{Z}}_f^{(t)(\ell)} = \hat{\mathbf{Z}}_f^{(t)(\ell)} + \sigma\left(\text{Softmax}\left(\text{LayerNorm}\left(\hat{\mathbf{Z}}_f^{(t)(\ell)}\right)^\top\right)\right)^\top$$

$$\psi_f^{(t)(\ell)} = \text{MEAN}\left(\tilde{\mathbf{Z}}_f^{(t)(\ell)} + \sigma\left(\text{LayerNorm}\left(\tilde{\mathbf{Z}}_f^{(t)(\ell)}\right)\mathbf{W}_P^{(1)}\right)\mathbf{W}_P^{(2)}\right)$$

where $\text{MEAN}(.)$ is mean function along the hyperedge dimension. The above pooling method uses a similar architecture as the feature extractor to capture both cross-hyperedge and cross-feature dependencies. However, there are two main differences: First, to reduce the number of parameters, we bind features in a non-parametric manner using a $\text{Softmax}(.)$ function. Second, the first line, the GRU cell [45], is used to update the encodings over time, capturing the dynamics of the functional systems.

④ FS-FS phase: Finally, we perform structural learning at the level of functional systems to capture the dependencies of brain activity at a higher level and use it to obtain brain-level encoding. To this end, we assume that all functional systems are connected and use a self-attention mechanism [43] to learn the strength of the dependencies of functional systems. Given $\psi_f^{(t)(\ell)}$ for each $f \in \mathcal{F}$, let $\Psi^{(t)(\ell)} \in \mathbb{R}^{|\mathcal{F}| \times d}$ be the matrix whose rows are $\psi_f^{(t)(\ell)}$. We use

$$\mathcal{A}_{\mathcal{F}}^{(t)} = \text{SIGMOID}\left(\frac{Q_{\mathcal{F}}^{(t)} K_{\mathcal{F}}^{(t)}}{\sqrt{K_{\mathcal{F}}^{(T)^\top}}}\right), \tag{7}$$

where $Q_{\mathcal{F}}^{(t)} = \Psi^{(t)(\ell)} \mathbf{W}_{\mathcal{F}}^{(1)}$, $K_{\mathcal{F}}^{(t)} = \Psi^{(t)(\ell)} \mathbf{W}_{\mathcal{F}}^{(2)}$, and $\mathbf{W}_{\mathcal{F}}^{(1)}$ and $\mathbf{W}_{\mathcal{F}}^{(2)}$ are learnable matrices. Accordingly, we can obtain the brain-level encoding as the weighted aggregation of functional system encodings $\Upsilon^{(t)(\ell)} = \sum_{f \in \mathcal{F}} \mathcal{A}_{f,f}^{(t)} \psi_f^{(t)(\ell)}$.

**Joint Training.** In the training phase, we want the model to learn to detect abnormal brain activities at the level of higher-order dependencies (i.e., hyperedges), ROIs (i.e., nodes), and function system or Brain. Accordingly, we use contrastive learning to train the model in an unsupervised manner. We generate negative samples at the level of hyperedges and nodes in the hypergraph. We adopt the commonly used negative sample generation method [25, 26] to generate negative hyperedges. That is, for each hyperedge $e = \{u_1, u_2, \ldots, u_k\}$ we choose $\alpha\%$ of nodes connected by $e$ and change them to a randomly selected set of vertices. We use binary cross-entropy loss, $\mathcal{L}_{\text{ENTROPY}}$, to learn hyperedge encodings. To learn the time series encoding, we follow existing studies [46, 47] and replace a brain signal in the time window with another signal that is randomly selected from the batch. We also follow these studies and use the contrastive loss, $\mathcal{L}_{\text{CONTRAST}}$ proposed by Woo et al. [47].

**Loss Function.** As we discussed earlier, ROIs in the same functional systems have similar patterns, and accordingly, it is expected to have similar encodings. To this end, inspired by DGI [48], we maximize the mutual information between ROIs encodings and their corresponding functional system encoding. We refer to this loss function as $\mathcal{L}_{\text{MI}}$. Accordingly, we aim to minimize $\mathcal{L} = \theta_1 \mathcal{L}_{\text{ENTROPY}} + \theta_2 \mathcal{L}_{\text{CONTRAST}} - (1 - \theta_1 - \theta_2) \mathcal{L}_{\text{MI}}$.

## 4 Experiments

**Datasets.** We use five real-world datasets: ① PD [49] consists of the functional MRI images of 25 participants with and 21 participants without PD, who do the ANT task [50]. ② ADHD [51] contains data for 100 subjects in the ADHD group and 100 subjects in the typically developed (TD) control group. ③ The Seizure detection TUH-EEG dataset [52] consists of EEG data (31 channels) of 642 subjects. ④ ASD [53] contains data for 45 subjects in the ASD group and 45 subjects in the TD control group. ⑤ ABIDE [54] consists of resting-state functional MRI of 1009 subjects (516 with ASD) parcellated by Craddock 200 atlas. For the first part of the experiment, we follow the

Table 1: Performance on anomaly detection: Mean AUC (%) $\pm$ standard deviation. Boldfaced letters shaded blue indicate the best result. N/A: The method has numerical precision or computational issues.

| Methods | PD | | ABIDE | | ADHD | | TUH-EEG | | ASD | |
|---|---|---|---|---|---|---|---|---|---|---|
| Anomaly % | 1% | 5% | 1% | 5% | 1% | 5% | 1% | 5% | 1% | 5% |
| **Hyperedge Anomaly Detection — Graph-based Methods** | | | | | | | | | | |
| GOUTLIER | $61.42_{\pm1.04}$ | $59.98_{\pm2.21}$ | $64.08_{\pm1.58}$ | $63.32_{\pm1.48}$ | $65.37_{\pm0.93}$ | $64.70_{\pm2.09}$ | $65.61_{\pm1.82}$ | $64.12_{\pm0.97}$ | $60.85_{\pm0.97}$ | $59.13_{\pm1.86}$ |
| NETWALK | $69.71_{\pm1.99}$ | $69.02_{\pm2.83}$ | $67.85_{\pm1.72}$ | $67.12_{\pm1.23}$ | $70.29_{\pm2.15}$ | $69.86_{\pm2.58}$ | $71.14_{\pm1.36}$ | $70.27_{\pm1.42}$ | $69.07_{\pm2.20}$ | $68.52_{\pm2.55}$ |
| BRAINGNN | $73.48_{\pm1.75}$ | $72.84_{\pm1.52}$ | $72.04_{\pm1.27}$ | $71.47_{\pm1.59}$ | $79.02_{\pm1.85}$ | $78.64_{\pm1.43}$ | $72.96_{\pm1.58}$ | $71.73_{\pm1.14}$ | $72.14_{\pm1.25}$ | $71.82_{\pm1.73}$ |
| BRAINNETCNN | $75.04_{\pm1.52}$ | $74.63_{\pm0.94}$ | $72.74_{\pm0.86}$ | $72.31_{\pm1.20}$ | $80.58_{\pm1.62}$ | $79.95_{\pm2.01}$ | $73.06_{\pm1.74}$ | $72.87_{\pm1.31}$ | $72.68_{\pm2.12}$ | $72.01_{\pm1.45}$ |
| BN-TRANSFORMER | $78.64_{\pm1.01}$ | $77.28_{\pm1.24}$ | $76.64_{\pm1.28}$ | $75.49_{\pm1.37}$ | $85.83_{\pm1.97}$ | $85.14_{\pm1.67}$ | $75.91_{\pm1.72}$ | $75.24_{\pm1.53}$ | $74.92_{\pm1.18}$ | $74.11_{\pm1.37}$ |
| ADMIRE | $81.36_{\pm1.73}$ | $80.98_{\pm2.05}$ | $77.26_{\pm1.43}$ | $76.58_{\pm1.19}$ | $86.23_{\pm1.74}$ | $85.18_{\pm2.21}$ | $76.68_{\pm1.82}$ | $75.14_{\pm1.67}$ | $86.52_{\pm1.72}$ | $85.44_{\pm1.49}$ |
| **Hypergraph-based Methods** | | | | | | | | | | |
| NHP | $73.24_{\pm0.84}$ | $72.65_{\pm1.06}$ | $69.32_{\pm1.81}$ | $68.79_{\pm1.12}$ | $82.61._{\pm0.64}$ | $81.22_{\pm1.34}$ | $73.07_{\pm1.49}$ | $72.24_{\pm0.67}$ | $67.98_{\pm0.37}$ | $67.17_{\pm1.41}$ |
| HYPERSAGCN | $75.82_{\pm1.64}$ | $74.10_{\pm1.28}$ | $72.52_{\pm0.73}$ | $71.06_{\pm1.18}$ | $84.22_{\pm1.61}$ | $83.96_{\pm1.47}$ | $73.99_{\pm0.83}$ | $72.65_{\pm0.97}$ | $73.26_{\pm1.08}$ | $73.18_{\pm0.92}$ |
| HADIB | $\mathbf{82.25_{\pm1.05}}$ | $\mathbf{81.12_{\pm1.14}}$ | $\mathbf{79.01_{\pm0.88}}$ | $\mathbf{78.99_{\pm1.07}}$ | $\mathbf{92.23_{\pm1.33}}$ | $\mathbf{90.87_{\pm1.16}}$ | $\mathbf{78.92_{\pm1.36}}$ | $\mathbf{78.04_{\pm1.05}}$ | $\mathbf{88.57_{\pm0.91}}$ | $\mathbf{89.45_{\pm1.08}}$ |
| **Time Series-based Methods** | | | | | | | | | | |
| USAD | $64.22_{\pm1.07}$ | $63.37_{\pm1.28}$ | $64.71_{\pm1.75}$ | $63.01_{\pm1.61}$ | $72.79_{\pm1.48}$ | $72.19_{\pm0.94}$ | $72.81_{\pm1.42}$ | $71.36_{\pm1.03}$ | $66.28_{\pm1.16}$ | $65.17_{\pm1.15}$ |
| MVTS | N/A | N/A | N/A | N/A | N/A | N/A | $80.99_{\pm1.36}$ | $80.27_{\pm1.49}$ | N/A | N/A |
| **ROI-level Anomaly Detection — Graph-based Methods** | | | | | | | | | | |
| GOUTLIER | $60.18_{\pm0.54}$ | $57.14_{\pm0.89}$ | $62.84_{\pm1.15}$ | $61.53_{\pm1.08}$ | $68.97_{\pm1.16}$ | $67.12_{\pm0.93}$ | $65.18_{\pm1.09}$ | $65.01_{\pm1.57}$ | $59.67_{\pm1.42}$ | $58.49_{\pm1.35}$ |
| NETWALK | $68.62_{\pm0.84}$ | $67.12_{\pm1.19}$ | $67.02_{\pm1.21}$ | $66.17_{\pm1.07}$ | $75.16_{\pm1.23}$ | $74.73_{\pm1.01}$ | $72.21_{\pm0.91}$ | $71.62_{\pm1.46}$ | $71.28_{\pm1.17}$ | $71.02_{\pm1.49}$ |
| BRAINGNN | $71.42_{\pm0.92}$ | $70.78_{\pm1.22}$ | $70.78_{\pm1.22}$ | $70.11_{\pm1.06}$ | $81.14_{\pm1.05}$ | $80.83_{\pm0.87}$ | $73.06_{\pm1.14}$ | $72.74_{\pm0.86}$ | $72.54_{\pm1.38}$ | $71.12_{\pm1.19}$ |
| BRAINNETCNN | $72.51_{\pm1.16}$ | $72.06_{\pm1.27}$ | $73.08_{\pm1.38}$ | $72.59_{\pm1.03}$ | $82.79_{\pm1.08}$ | $81.12_{\pm1.16}$ | $73.98_{\pm1.24}$ | $73.01_{\pm1.08}$ | $73.18_{\pm0.95}$ | $72.88_{\pm1.04}$ |
| BN-TRANSFORMER | $73.08_{\pm1.07}$ | $72.53_{\pm1.12}$ | $76.72_{\pm1.13}$ | $76.04_{\pm1.06}$ | $86.13_{\pm1.21}$ | $86.11_{\pm1.82}$ | $77.96_{\pm1.32}$ | $77.08_{\pm1.06}$ | $76.05_{\pm1.52}$ | $75.72_{\pm1.18}$ |
| **Hypergraph-based Methods** | | | | | | | | | | |
| NHP | $70.73_{\pm1.08}$ | $69.37_{\pm1.17}$ | $70.88_{\pm1.14}$ | $70.21_{\pm0.96}$ | $83.01_{\pm1.03}$ | $82.14_{\pm1.10}$ | $73.62_{\pm1.14}$ | $73.01_{\pm0.92}$ | $72.25_{\pm0.88}$ | $71.07_{\pm1.06}$ |
| HYPERSAGCN | $72.81_{\pm1.15}$ | $71.97_{\pm1.08}$ | $73.36_{\pm1.27}$ | $72.65_{\pm1.16}$ | $83.94_{\pm1.13}$ | $83.01_{\pm0.92}$ | $75.62_{\pm1.12}$ | $74.83_{\pm0.78}$ | $74.93_{\pm1.47}$ | $74.15_{\pm1.19}$ |
| HASHNWALK | $72.48_{\pm1.01}$ | $72.03_{\pm1.47}$ | $71.98_{\pm1.07}$ | $70.96_{\pm1.12}$ | $83.45_{\pm1.03}$ | $83.07_{\pm0.91}$ | $74.48_{\pm1.15}$ | $73.49_{\pm0.96}$ | $73.98_{\pm0.73}$ | $73.14_{\pm1.04}$ |
| HADIB | $\mathbf{79.14_{\pm1.12}}$ | $\mathbf{78.57_{\pm0.97}}$ | $\mathbf{80.26_{\pm1.62}}$ | $\mathbf{79.87_{\pm1.83}}$ | $\mathbf{92.34_{\pm0.97}}$ | $\mathbf{92.01_{\pm1.56}}$ | $\mathbf{82.98_{\pm1.14}}$ | $\mathbf{82.10_{\pm1.09}}$ | $\mathbf{81.72_{\pm1.05}}$ | $\mathbf{80.84_{\pm1.71}}$ |
| **Time Series-based Methods** | | | | | | | | | | |
| USAD | $63.27_{\pm1.66}$ | $62.75_{\pm1.38}$ | $66.29_{\pm1.24}$ | $66.08_{\pm1.13}$ | $82.87_{\pm2.03}$ | $80.52_{\pm1.84}$ | $72.03_{\pm1.17}$ | $71.48_{\pm1.05}$ | $71.62_{\pm1.58}$ | $70.98_{\pm1.41}$ |
| MVTS | N/A | N/A | N/A | N/A | N/A | N/A | $83.53_{\pm1.91}$ | $82.41_{\pm1.02}$ | N/A | N/A |
| **Brain-level Anomaly Detection — Graph-based Methods** | | | | | | | | | | |
| NETWALK | $62.12_{\pm2.10}$ | $61.54_{\pm1.49}$ | $66.01_{\pm1.26}$ | $65.12_{\pm1.74}$ | $83.11_{\pm1.02}$ | $82.81_{\pm1.61}$ | $71.06_{\pm1.05}$ | $69.94_{\pm1.12}$ | $72.85_{\pm1.17}$ | $72.21_{\pm1.34}$ |
| BRAINGNN | $67.69_{\pm1.14}$ | $67.02_{\pm1.45}$ | $69.62_{\pm6.04}$ | $69.03_{\pm5.96}$ | $84.59_{\pm1.26}$ | $83.72_{\pm1.35}$ | $72.41_{\pm1.38}$ | $71.55_{\pm1.16}$ | $75.12_{\pm1.33}$ | $74.57_{\pm1.52}$ |
| BRAINNETCNN | $68.32_{\pm0.86}$ | $67.24_{\pm1.08}$ | $73.02_{\pm5.04}$ | $72.95_{\pm5.47}$ | $85.84_{\pm0.96}$ | $85.07_{\pm1.52}$ | $73.92_{\pm0.97}$ | $73.07_{\pm1.51}$ | $75.96_{\pm1.66}$ | $75.03_{\pm1.28}$ |
| BN-TRANSFORMER | $70.22_{\pm0.97}$ | $68.14_{\pm1.52}$ | $76.01_{\pm3.22}$ | $75.68_{\pm3.02}$ | $87.54_{\pm1.04}$ | $86.92_{\pm1.48}$ | $79.36_{\pm1.71}$ | $78.08_{\pm1.16}$ | $77.19_{\pm2.01}$ | $76.58_{\pm1.73}$ |
| **Hypergraph-based Methods** | | | | | | | | | | |
| NHP | $67.74_{\pm1.13}$ | $67.55_{\pm0.91}$ | $70.86_{\pm1.42}$ | $69.13_{\pm0.99}$ | $84.12_{\pm1.27}$ | $84.01_{\pm1.35}$ | $74.42_{\pm1.04}$ | $74.06_{\pm1.22}$ | $75.05_{\pm1.16}$ | $74.63_{\pm1.43}$ |
| HYPERSAGCN | $68.97_{\pm1.16}$ | $68.04_{\pm0.96}$ | $74.06_{\pm1.77}$ | $73.46_{\pm1.48}$ | $86.94_{\pm1.63}$ | $86.17_{\pm1.49}$ | $75.31_{\pm0.85}$ | $74.79_{\pm1.09}$ | $76.72_{\pm1.32}$ | $75.81_{\pm1.58}$ |
| HADIB | $\mathbf{76.14_{\pm1.58}}$ | $\mathbf{75.82_{\pm1.37}}$ | $\mathbf{81.08_{\pm0.87}}$ | $\mathbf{80.02_{\pm1.07}}$ | $\mathbf{93.52_{\pm0.98}}$ | $\mathbf{92.45_{\pm1.10}}$ | $\mathbf{83.65_{\pm1.87}}$ | $\mathbf{82.54_{\pm1.07}}$ | $\mathbf{82.41_{\pm1.92}}$ | $\mathbf{81.76_{\pm1.83}}$ |

Table 2: Ablation study on HADIB. AUC scores on hyperedge anomaly detection.

| | Methods | PD | ADHD | ASD |
|---|---|---|---|---|
| | HADIB | $81.12_{\pm1.14}$ | $90.87_{\pm1.16}$ | $89.45_{\pm1.08}$ |
| ① | Remove Feature Extraction | $76.52_{\pm2.98}$ | $84.80_{\pm1.53}$ | $70.09_{\pm0.94}$ |
| ② | Remove Time Scales | $79.31_{\pm0.85}$ | $85.29_{\pm0.97}$ | $71.23_{\pm1.19}$ |
| ③ | Remove Self-attention | $77.41_{\pm1.05}$ | $85.98_{\pm1.26}$ | $70.68_{\pm1.47}$ |
| ④ | Remove Gru | $80.79_{\pm0.43}$ | $88.23_{\pm1.07}$ | $76.08_{\pm1.25}$ |
| ⑤ | Remove $\mathcal{L}_{\text{ENTROPY}}$ | $58.12_{\pm5.89}$ | $57.31_{\pm8.03}$ | $58.18_{\pm0.98}$ |
| ⑥ | Remove $\mathcal{L}_{\text{CONTRAST}}$ | $59.50_{\pm7.93}$ | $72.53_{\pm1.16}$ | $68.44_{\pm0.96}$ |
| ⑦ | Remove $\mathcal{L}_{\text{MI}}$ | $65.17_{\pm3.97}$ | $80.74_{\pm1.08}$ | $69.93_{\pm1.27}$ |
| ⑧ | Use Simple Message-passing | $77.67_{\pm1.09}$ | $84.32_{\pm1.18}$ | $74.97_{\pm1.13}$ |
| ⑨ | Use Pair-wise Graph | $77.64_{\pm0.38}$ | $86.94_{\pm1.20}$ | $74.13_{\pm1.12}$ |

methodology used in existing studies [26, 55, 39] and inject 1% and 5% anomalous edges into the brain networks in the control group.

**Baselines.** We compare our HADIB with state-of-the-art methods. In all tasks, we use ① Graph-based methods: GOutlier [56], NETWALK [57], BRAINGNN [58], BRAINNETCNN [59], ADMIRE [39], and BNTRANSFORMER [9]. ② Hypergraph-based methods: HYPERSAGCN [60], NHP [61], HASHNWALK [29]. ③ Time-series-based methods: USAD [62] and MVTS [63]. We may exclude some baselines in some tasks as they cannot be applied in that setting.

**Quantitative Results.** In the first experiment, we compare the performance of HADIB with baselines in hyperedge, ROI, and graph anomaly detection tasks. Table 1 reports the the AUC of HADIB and baselines. HADIB outperforms all the baselines in all three tasks. The reason is fourfold: ① Graph-based methods can only capture the pair-wise dependencies of ROIs, missing higher-order dependencies. ② Hypergraph-based methods also miss the temporal properties as well as the dynamics of the functional connectivity. ③ HADIB's training strategy and encoding of the brain at different level of granularity can provide complementary information. ④ HADIB is an end-to-end model that simultaneously learns the hypergraph structure and its element (e.g., node, hyperedge)

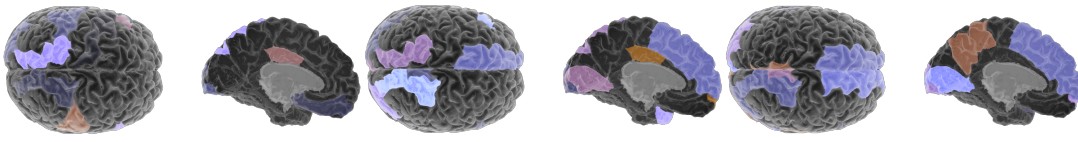

| (a) Results on PD | (b) Results on ADHD | (c) Results on ASD |

Figure 2: The distribution of found anomalies by HADIB in condition groups.

encodings. However, all the baselines use pre-computed functional connectivity, which limits their generalizability.

**Ablation Study.** We conduct ablation studies to validate the effectiveness of HADIB's critical components. Table 2 shows the AUC for the hyperedge anomaly detection task. The first row reports the performance of the complete HADIB implementation. Each subsequent row shows results for HADIB with one module modification: row 1 replaces feature extraction with Pearson's correlation, row 2 considers only one time scale, row 3 removes the self-attention mechanism, row 4 removes GRU cell in the pooling, rows 5-7 removes one of the loss functions at each time, rows 8 replaces the tetra stage messaging with a simple message passing, and finally, the last row converts the hypergraph to a graph. These results show that each component is critical for achieving HADIB's superior performance. The greatest contribution comes from the training process and loss functions, followed by Feature Extraction modules.

## 5 Case Studies

In the following case studies, we train our model on the healthy control group and then test it on the condition group. To this end, we report how anomalous brain activities found by HADIB are distributed in the brains of people living with PD/ADHD/ASD.

**Parkinson's Disease.** Figure 2 reports the distribution of anomalous ROIs within the PD group. A majority (95%) of the identified anomalies by HADIB involve ROIs located in one of the Left Thalamus, Supramarginal Gyrus, Superior Parietal, Medial Orbitofrontal, or Pars Opercularis. Interestingly, these ROIs are correlated with some PD symptoms (e.g., affected motor skills). Also, these results are consistent with previous studies using resting-state fMRI [64, 65].

**Attention Deficit Hyperactivity Disorder.** Figure 2 reports the distribution of anomalous ROIs within the brain networks of the ADHD group. Most found abnormal ROIs (80% of all found anomalies) by HADIB are located in the Left Temporal Pole, Frontal Pole, Left Lateral Occipital, and Lingual Gyrus. These findings are consistent with previous studies on ADHD by using diffusion tensor imaging [66] and Forman–Ricci curvature changes [4].

**Autism Spectrum Disorder.** The distribution of anomalous ROIs within the brain networks of the ASD group is visualized in Figure 2. More than 80% of all found anomalies by HADIB involve ROIs located in the Right Cerebellum Cortex, Left Cerebellum Cortex, Right Cerebellum Cortex, Frontal Pole, Left Lateral Occipital, and Right Superior Temporal Gyrus. Our results on findings of abnormal activity in the cerebellum cortex are consistent with previous studies [67].

## 6 Conclusion

We present HADIB, an end-to-end unsupervised learning method on brain time series data to detect abnormal brain activity at different levels of granularity that might suggest a brain disease or disorder. HADIB uses a hierarchical multivariate time-series encoder to capture both cross-time and cross-ROI dependencies of ROIs' signals at different time scales and employs a temporal hypergraph constructor module on top of that to learn the temporal higher-order dependencies between ROIs. Using a novel hierarchical tetra-stage message passing on the constructed hypergraph, HADIB first learns ROI-level, hyperedge-level, and brain-level encodings and then leverages them to detect abnormal brain activity at different levels. Our experimental results show the superior performance of HADIB against baselines and the potential of HADIB in detecting abnormal brain activity.

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
