# OpenReview forum: "Learning Temporal Higher-order Patterns to Detect Anomalous Brain Activity"
_NeurIPS.cc/2023/Workshop/AI4Science — NeurIPS2023-AI4Science Poster_

### Official Review · Reviewer_oebi · 2023-10-22
**Hypergraph method for detecting anomalous brain activity**

**Rating:** 6
**Confidence:** 4

**Review:**

The paper presents a useful method based on hypergraph learning. Overall, it is an intriguing work, but needs further elaboration on specific methodology and implications of the findings. Here are some review comments regarding this paper.

1)	This paper could be improved with more specific details on the four-stage message-passing mechanism and the adaptive hypergraph pooling, as these are key components of the solution.
2)	As the major applications of HADiB is focused on detecting anomalous brain activity, it would be helpful to provide a bit more context on the results or findings related to these specific applications. More discussions on its potential application in other related tissue types would be helpful too.
3)	It would be beneficial to mention the potential generalizability of HADiB model to other neurological disorders or applications. Additionally, consider including a sentence or two on future work or next steps in this research, such as potential improvements or extensions of the HADIB model.

---

### Meta-Review · Area_Chair_AFkv · 2023-10-28

**Recommendation:** Accept (Poster)
**Confidence:** 4

**Metareview:**

As indicated by the reviewer, the paper presents an interesting methodology for anomaly detection involving the construction of dynamic hypergraph representations of functional connectivity (either EEG or fMRI). Extensive experimentation has been conducted against a variety of graph neural network and hyper graph neural network baselines, which this model seems to outperform on several datasets of different neurological disorders.

 It would be interesting to see this work presented and discussed formally in the workshop. Meanwhile, I would encourage the authors to also consider the following points in a subsequent version.

POINTS NEEDING CLARIFICATION:

1. The adjacency matrix as defined in Eq. (2) seems to be a bivariate measure, making it unclear how higher order relations are captured to construct the hypergraph- please correct me if this is not how the equation is supposed to be interpreted.
2. It is noteworthy that rather than disease identification, this work focuses on the problem of anomaly detection. The datasets used in this work were not assembled with this goal in mind. Therefore, the authors seem to have simulated the presence of anomalous connections in the graphs. It would be good to explain this simulation in the appendix, since it is highly relevant to the evaluation. Additionally, since the datasets were assembled for discriminating between two different sets of diagnostic populations (i.e. FC graphs should sampled from distinct population graph distributions), does this affect the semantics of normal vs anomalous edges?
3. It would also be nice to formalize the loss functions in the appendix and clarify how hyper parameters were selected for the experiments.